# Understanding Vasomotion of Lung Microcirculation by In Vivo Imaging

**DOI:** 10.3390/jimaging5020022

**Published:** 2019-01-22

**Authors:** Enrico Mazzuca, Andrea Aliverti, Giuseppe Miserocchi

**Affiliations:** 1Department of Electronics, Information and Bioengineering, Politecnico di Milano, 20133 Milan, Italy; 2Department of Experimental Medicine, Università di Milano Bicocca, 20900 Milan, Italy

**Keywords:** in vivo microscopy, lung capillaries, edema, image-based modeling

## Abstract

The balance of lung extravascular water depends upon the control of blood flow in the alveolar distribution vessels that feed downstream two districts placed in parallel, the corner vessels and the alveolar septal network. The occurrence of an edemagenic condition appears critical as an increase in extravascular water endangers the thinness of the air–blood barrier, thus negatively affecting the diffusive capacity of the lung. We exposed anesthetized rabbits to an edemagenic factor (12% hypoxia) for 120 min and followed by in vivo imaging the micro-vascular morphology through a “pleural window” using a stereo microscope at a magnification of 15× (resolution of 7.2 μm). We measured the change in diameter of distribution vessels (50–200 μm) and corner vessels (<50 μm). On average, hypoxia caused a significant decrease in diameter of both smaller distribution vessels (about ~50%) and corner vessels (about ~25%) at 30 min. After 120 min, reperfusion occurred. Regional differences in perivascular interstitial volume were observed and could be correlated with differences in blood flow control. To understand such difference, we modelled imaged alveolar capillary units, obtained by Voronoi method, integrating microvascular pressure parameters with capillary filtration. Results of the analysis suggested that at 120 min, alveolar blood flow was diverted to the corner vessels in larger alveoli, which were found also to undergo a greater filtration indicating greater proneness to develop lung edema.

## 1. Introduction

The air–blood barrier (ABB) in the gas diffusion compartment of the lung assures efficient gas exchanges, thanks to its extreme thinness (0.2–0.5 μm). Such thinness is maintained by a specific arrangement of the macromolecular extravascular structure coupled to a tight control on the amount of extravascular water, so that the lung appears to be inherently very resistant to the development of edema [1]. Yet, a remarkable increase in microvascular filtration (termed as edemagenic condition) may be generated by the combination of various factors: an increase in blood pressure or flow and/or of water microvascular permeability due to the degradation of the native biomechanical properties of the macromolecular mesh of the capillary walls and of the perivascular interstitial space [1,2,3]. The occurrence of an edemagenic condition in the lung appears critical as an increase in extravascular water endangers the thinness of the ABB, thus negatively affecting the diffusive capacity of the lung. We exposed experimental animals to hypoxia, a well-known edemagenic factor, characterized by a variable and complex matching of all the factors mentioned above [1,2,3]. It is also well known that on hypoxia exposure, a considerable rearrangement of the blood flow distribution occurs in the lung, leading to pulmonary hypertension of various degree due to an increase of peripheral vascular resistances. The latter was attributed to vasoconstriction of arterioles in the range of 150–200 μm [4] and indications are that vasoconstriction mostly occurred in lung regions where initial edema was developing [5]. Yet, no data were so far available concerning the most distant districts of the alveolar microcirculation that includes the alveolar distribution vessels that feed downstream two distinct districts placed in parallel, namely the corner vessels, running at the edge of the alveoli, and the extended septal capillary network distributed on the alveolar surface, mostly involved in gas diffusion.

The aim of this study was to apply an in vivo imaging technique to gather data relative to the more distant vascular districts that are obviously mostly involved in microvascular fluid exchange. We took advantage of a recently developed technique allowing us to view directly [6] the subpleural alveolar district through a “pleural window” leaving the pleural sac intact. The imaging data were then used to implement the alveolar–capillary topology, as suggested by a theoretical model [7] in order to derive semi-quantitative estimates concerning the role of vasomotion in the control of blood flow and microvascular filtration at the level of the septal circulation; this district cannot be imaged directly and is most involved in the gas diffusion process.

## 2. Materials and Methods

### 2.1. Experimental Setup

The ethical approval for the experimental procedures used in our research activity was obtained from Milano-Bicocca University Ethical Committee (University Milano Bicocca IR2013/1). Further, animal experimentation was performed according to the Helsinki Convention for the use and care of animals. Four adult male New Zealand White rabbits (weight range 1–1.5 kg) were anesthetized, tracheotomized, paralyzed, and mechanically ventilated. Heart rate was continuously monitored to allow the administration of supplementary anaesthesia when it increased by more than 15%.

The skin and external intercostal muscles on the right side of the chest were resected down to the endothoracic fascia to clear a surface area of about 0.5 cm^2^. Using fine forceps under stereomicroscopic view, the endothoracic fascia was carefully stripped to open a “pleural window”, about 20 mm^2^ large, allowing a neat view of the morphology of the underlying alveoli and microvessels thanks to the transparency of the parietal pleura [3,6,8]. The great advantage of this technique is it preserves the integrity of the pleural compartment that assures the mechanical coupling between lung and chest wall [4]. The window was opened in the seventh intercostal space which allowed to visualize at end expiration the highly vascularised lobar margin of the lower lobe, which displays an alveolar field including about 100 alveoli, the corner vessels delimiting the alveolar contour, and the distribution vessels.

Images were acquired using a SMZ stereo microscope (Nikon, Shinagawa, Japan), equipped with a CMOS camera (OPTIKAM-B5) (Optika Microscopes, Ponteranica, Italy). A magnification of 15× was used allowing a resolution of 7.2 μm; this magnification was chosen as a compromise between sufficient resolution and a field of view large enough to track the same subpleural region throughout the experiment. An LED ring-light illuminator was anchored to microscopic optics to provide a uniform lighting of the alveolar field (Figure 1).

After baseline imaging of subpleural micro-vascular and alveolar morphology during air breathing, rabbits were exposed to a hypoxic mixture (12% oxygen and nitrogen) delivered along the inspiratory line of a Y intratracheal tube equipped with an inspiratory–expiratory valve. Images were then acquired at end expiration every 10 min in the first hour and every 20 min for the following 60 min. During the experiment, the same subpleural regions (one for each rabbit) could be imaged, so that the same alveolar units and the same corner and distribution vessels could be tracked. Tidal volume of mechanical ventilation was set as to provide the same peak inspiratory alveolar pressure throughout the whole experiment (12 cmH_2_O).

At the end of the experiment, animals were euthanized by anaesthetic overdose, thorax was opened, and some lung samples were cut and weighted. They were desiccated in oven at 60° for 2 days and then the dry weight was determined to obtain the wet-to-dry ratio.

### 2.2. Image Analysis

Image analysis was performed in control conditions and at different times during hypoxia exposure. Only images at end expiration have been considered (Figure 2A).

#### 2.2.1. Segmentation of Vessels:

We applied the method developed by [9] successfully used to study lung microvascular geometry in developing lung edema [8]. The method defines mathematically the border between vessels and perivascular interstitial space relying on a semiautomatic procedure that identifies the sharp change of the moving average of the grey level shifting from the inside of the vessel towards the adjacent interstitial region. The method was validated by measuring objects of known size and by varying the ratio of grey level among the objects. Accordingly, the segmentation process was carried using greyscale images.

Two different districts of microvessels could be imaged:Distribution vessels; two groups were considered: one with diameter ranging 250–100 μm, the other with diameter ranging 50–100 μm.Corner vessels, clearly visible as running along the edges of the alveoli, having a diameter ranging from 10 to 50 μm.

Septal vessels, ranging in diameter from 5 to 8 μm, could not be directly visualized.

Based on [9], the transition between the vessel and the interstitial space was identified by a sharp change in local image intensity corresponding to the vessel wall. The average diameter of both distribution and corner vessels was estimated by averaging 4 vessel lumen width, drawn manually perpendicular to vessel direction, as shown in Figure 2B.

#### 2.2.2. Estimate of interstitial space volume:

Regions of interest (ROI) containing on the average 8–10 alveoli were identified over the lung surface. Alveolar units were manually segmented, by following their borders identified by the grey level between air and tissue phase (Figure 2C). The difference between overall area encompassing the ROI and the alveolar area was considered as representative of peri-alveolar interstitial space [9].

### 2.3. Building a 2D Morpohological Model of the Alveolar Capillary Circulation

The alveolar capillary unit (ACU) is defined as the microvascular compartment including the distribution vessels and the two districts in parallel, namely the corner vessels and the septal flow (Figure 3A). Figure 3B shows a 2D image-based model obtained by applying a 2D constrained Voronoi [7] to a defined ROI taken from an experimental image. Corner vessels junctions were identified; to build the septal capillary network, we chose a random distribution of N Voronoi points within the ACU. The value of N was chosen so as to match the morphological constraint of the ratio between capillary and interstitial alveolar volume equal to 1.54, a value provided for rabbits [10] corresponding to the maximum extension of a pulmonary capillary network. Other details of the morphological model can be found in the Appendix A. Inlet points of the ACU were chosen as the terminal part of visible subpleural arterioles, while venous outlets were assumed to be on the opposite site of the inlet points in the 2D network. Input and output points was chosen as to provide full perfusion of the septal circulation in baseline condition.

In summary, for each ACU, the following dataset was defined: (1) coordinated of nodes, (2) inlet and outlet points, and (3) corner vessels diameters, which were measured by hand for three representative time instants: baseline, 30 min, and 120 min.

### 2.4. Statistical Analysis

One-way repeated measurement analysis of variance (RM-ANOVA) with independent variable being time was applied on the absolute values of arteriolar diameters to estimate the dependence of vasoactive response of distribution microvessels on time of hypoxia exposure. The dependence of corner radius on time of hypoxia exposure was evaluated by a one-way RM-ANOVA, with independent variable being time. Statistical analysis was performed by SigmaStat software v11.0 (San Jose, CA, USA).

## 3. Results

### 3.1. Wet-to-Dry Ratio and Blood Gas

Wet-to-dry ratio of the lung in control rabbits was 4.3 ± 0.72; after hypoxia exposure, it increased to 4.91 ± 0.14 (*p* = 0.092). A 14% increase in wet-to-dry ratio is compatible with a condition defined as “interstitial” edema [3]. Arterial PO_2_ was 88 ± 2 mmHg in control conditions; after 30 min of hypoxia, the corresponding value was 36 ± 16 mmHg.

### 3.2. Imaging Data

Figure 4A shows no appreciable change in distribution vessel diameter under a 1-h steady condition (*p* = 0.42). During hypoxia exposure, the diameter of distribution vessels with diameter >100 μm (Figure 4B) remained essentially unchanged up to 120 min of observation; a significant difference was found only among 30 and 100 min (*p* = 0.001). For distribution vessels with diameter <100 μm (Figure 4C), two different patterns were observed: either no change, or remarkable decrease in diameter approaching complete closure after 30 min of hypoxia exposure; after this time, the diameter returned towards control at about 80 min, remaining thereafter essentially steady. On average, considering all the vessels in this domain, a significant decrease in diameter down to about 50% of control was observed at 30 min, followed by a return to control values at 80 min. The following statistically significant differences were found: baseline versus 10, 20, 30, and 40 min; 80, 100, and 120 min versus 10, 20, 30, 40, and 50 min (all differences having a *p*-value < 0.001).

Figure 5 presents data from two ROIs allowing to relate changes in alveolar interstitial fluid balance to microvessels vasomotion. The chosen ROIs are representative of a difference in fluid accumulation in the peri-microvascular interstitial space. On the left (Figure 5A) is the case of an increase in thickness of the peri-microvascular space, revealing the development of interstitial lung edema on hypoxia exposure. The corresponding changes in diameter of the distribution vessels, with diameter less than 100 μm (Figure 5C), showed either no change or complete closure, while the diameter of corner vessels increased. Figure 5B exemplifies the case where no perturbation in alveolar interstitial fluid balance occurred as the thickness of the interstitial space remained unchanged; a modest decrease in diameter of distribution vessels (less than 100 μm in diameter) was observed while the diameter of corner vessels steadily increased (Figure 5D).

Figure 6 presents the overall average behaviour of 120 corner vessels showing a mild decrease in diameter at 30 min (*p* < 0.05) and a progressive increase up to 120 min (*p* < 0.05).

## 4. Discussion

### 4.1. In Vivo Subpleural Microscopy

*Advantages*. Previous attempts of imaging of subpleural alveolar compartment largely interfered with unrestrained alveolar movement [11,12,13]. Our method offers the advantage to allow unrestrained movement of subpleural units, in a physiological condition of lung expansion, that preserves the integrity of the respiratory system and of microcirculation. Another advantage is the possibility to follow the same alveolar- capillary units during the experiment.

*Drawbacks*. Potential limitation of the study is represented by the fact that only subpleural units were studied and the discussion concerning their mechanical behavior does not allow us to extrapolate to the rest of the parenchyma. Moreover, the microvessel caliper estimated from imaging data might be affected by image artifact due to focal depth, although the visceral pleura is transparent and its thickness ranges as low as 10 μm.

### 4.2. The Current State of Knowledge on Pulmonary Vascular Response to Hypoxia

There is a general consensus on the fact that hypoxia leads to pulmonary hypertension reflecting an increase in peripheral vascular resistances. Arteriolar vasoconstriction was documented [14] as well as vascular remodeling leading to deposition of smooth muscles at arteriolar level [15]. Hypoxia is a clinical consequence of cardio-pulmonary disorders and pulmonary hypertension is considered as a vascular dysfunction relating to the biochemical and cellular control of vasomotion [16]. It is well known that hypoxia following altitude exposure exacerbates this dysfunction. Severe pulmonary hypertension may develop in low landers exposed to high altitude, and further, the degree of hypertension increases with the proneness to develop high altitude lung edema [17]. There is also evidence of blood flow redistribution in the human lung when edema develops [18]. The fact that no pulmonary hypertension is observed in Tibetan highlanders who appear to be very resistant to lung edema [17] stirs the debate on the possible significance of the inter-individual differences in the vascular adaptive response of the lung to hypoxia exposure.

A recent line of interpretation from a research group in Milano pivots around the perturbation induced by hypoxia on the microvascular–interstitial fluid dynamics and its consequences on the capillary–lung tissue mechanical interaction [19]. Pulmonary interstitial pressure in the peri-microvascular interstitial space was found to increase remarkably from control value of −10 cmH_2_O to about 5 cmH_2_O following an increase in microvascular filtration in hypoxic edemagenic condition [3]. Based on this finding, a theoretical model of alveolar micro-circulation [7] suggested that the increase in tissue pressure during developing edema would lead to two interconnected events: progressive closure of alveolar septal capillaries and corresponding increase in blood flow in the corner vessels that are placed in parallel with the septal ones. One should note that corner vessels, unlike septal capillaries, are prevented from closing when interstitial pressure increases, being kept patent by the pulling action of the elastic parenchymal forces. The shift of blood flow from septal network to corner vessels was interpreted as being preventive of a progression towards severe edema in the most delicate portion of the lung.

### 4.3. Different Behaviour of Alveolar Microvessels in Edemagenic Condition

This study allows to unveil some aspects of the control of blood flow in lung microcirculation, particularly in hypoxia-dependent edemagenic condition: data indicate that the upstream segment of alveolar lung microcirculation, namely the distribution vessels, being devoid of smooth muscles, can control downstream the flow to the ACUs.

Although distribution vessels >100 μm do not seem to play any role, vasoconstriction of distribution vessels <100 μm can reduce or even shut down blood flow to the ACUs after 30 min of hypoxia exposure. It is important to recall that unperfused capillary segments may provide fluid re-absorption from the interstitial compartment due to a remarkable decrease in capillary pressure and reversalof the transmural Starling pressure gradient [20]. Subsequent reopening occurred restoring downstream blood flow at 120 min. We provide the important indication (Figure 4 left) that vasoconstriction of distribution vessels occurs in ACUs that appear to be more prone to develop interstitial edema, as suggested by the increase in interstitial space thickness (Figure 4 left). In these same regions, the ~60% increase of corner diameter suggests re-direction of blood flow from the septal alveolar network to the corner district. This blood shift accounts for an increase in peripheral vascular resistances due to the decrease in vascular bed considering that the extension of the corner vessels is only a 15% of the septal circulation. It appears logical to admit that the larger this shift is as a consequence of a more extended edema formation, the greater will be the increase in pulmonary artery pressure.

In regions where no interstitial edema developed (Figure 4 right), no appreciable change in distribution vessels < 100 μm was observed and corner vessel diameter almost doubled. As corner and septal vessels have no smooth muscles, an increase in septal flow should have paralleled the increase in corner flow. Septal capillary recruitment has been reported on hypoxia exposure [21]: given the regional variability in the control of alveolar microcirculation, we believe this finding can only apply to non-edematous regions.

### 4.4. Image-Based Modeling

Based on a theoretical model previously presented [6], we attempt to derive indications on the role of vasomotion to control blood flow in the alveolar compartment as well as its relationship with interstitial fluid balance. The analysis was done on network models for baseline, 30 and 120 min of hypoxia exposure as reference timepoints. Two different networks from the same rabbit were considered and the hemodynamic parameters were estimated by applying the contour conditions presented in Table 1 to the ACU geometry. According to Reference [4], with edema progression, input arteriolar pressure *P_art_* does not change for 50-µm large arterioles, while some venular venoconstriction (increase in *P_ven_*) can be observed, as proposed also in Reference [22] for hypoxia-induced edema. Also, an increase in peri-microvascular interstitial pressure $P_{i}$ is foreseen from a nominal value of −10 cmH_2_O up to 3 cmH_2_O [3]. Table 1 reports input pressure parameters at the representative time-points.

Figure 7 shows the models of perfusion patterns for two ACU networks that were chosen based on a different baseline topology (Table 2): network 1 has larger alveoli (as indicated by longer corner vessels), thus a greater extension of the septal network, four times larger septal-to-corner flow ratio and a greater microvascular filtration flow. The assumption about a complete perfusion of the whole ACU in baseline condition allows the evaluation of the impact of the edemagenic perturbation on microvascular blood flows.

Table 2 summarizes the results of the analysis for the two networks relative to blood flow in the corner vessels and septal network (the two districts placed in parallel) as well as microvascular filtration flow in the septal regions (details of the analysis in the Appendix A and in Reference [7]).

For the same input parameters (Table 1), at 30 min of hypoxia exposure, a similar reduction in ACU blood flow was observed in both networks; yet, the septal-to-corner ratio remained essentially unchanged in ACU network 1 while it increased remarkably in ACU network 2 revealing a relative increase in septal flow. Please note that at 120 min, for ACU 1, blood flow was restored with respect to 30 min, even if from Figure 7 a greater de-recruitment is shown; the reason is that ACU flow in network 1 was compromised in a large part of the network, but the lower resistance in the upper right part largely contributes to the higher flow.

Filtration flow can be computed by entering the adequate parameters into the Starling Law [7]: the model allows to determine the hydraulic pressure along the ACU capillaries as well as the filtration surface. At baseline, the latter is greater in ACU 1 due to a greater extension of the septal network. Over time, the overall filtration surface decreases more in the larger ACU, reflecting the mechanical compression of the capillaries due to the reduction of transmural pressure.

Entering the same microvascular permeability in the Starling Law and normalizing filtration to capillary surface allows to focus on the difference of the two networks concerning the control of interstitial fluid balance. In network 1, filtration flow remains high, and increases by 45% at 120 min, despite some decrease in filtration surface. One can deduce that the proneness to develop interstitial edema is greater in ACU 1, and therefore the assumption of the same permeability as for ACU 2 leads to an underestimate of the real filtration flow in ACU 1.

Giving the heterogeneous distribution of blood flow in ACU 1 (Figure 7), the increase in filtration should be also heterogeneously distributed. Also, filtration in ACU 2 appears heterogeneously distributed, but is kept lower at any time point relative to ACU 1 and only increases by 30% after 120 min. In a previous study [23], we reported considerable heterogeneity in alveolar mechanics reflecting heterogeneity in alveolar size; yet, the same study showed that a substantially homogenous mechanical behavior of the lung resulted from homogenous distribution of these heterogeneities. The same principle could be extended to the alveolar peripheral vasomotion to control lung fluid balance; in fact, it appears tempting to consider blood flow diversion from regions being less resistant to edema to more resistant ones, as an intrinsic mechanism to preserve the efficiency of the diffusion/perfusion function.

## 5. Conclusions

The present study investigates the short-term and mid-term adaptive response of pulmonary micro-circulation in the subpleural region in a closed-chest rabbit lung during development of interstitial edema, following hypoxia exposure. The application of a theoretical model to the experimental images allows to estimate hemodynamic parameters from topological analysis and to address questions about the physiological mechanisms underlying the adaptive vascular response to an edemagenic condition. The study suggests that large alveolar units appear less efficient to counteract edema formation and the vasoactive response may in fact be regarded as a mechanism aimed at preventing the aggravation of developing edema at alveolar level, rather than as a dysfunction. It remains to be investigated which morpho-functional features are at the base of a greater proneness to develop alveolar edema. Indirect indications suggest that in humans the morpho-functional arrangement of the perfusion-ventilation coupling the alveolar units may be implicated [24].

## Figures and Tables

**Figure 1 jimaging-05-00022-f001:**
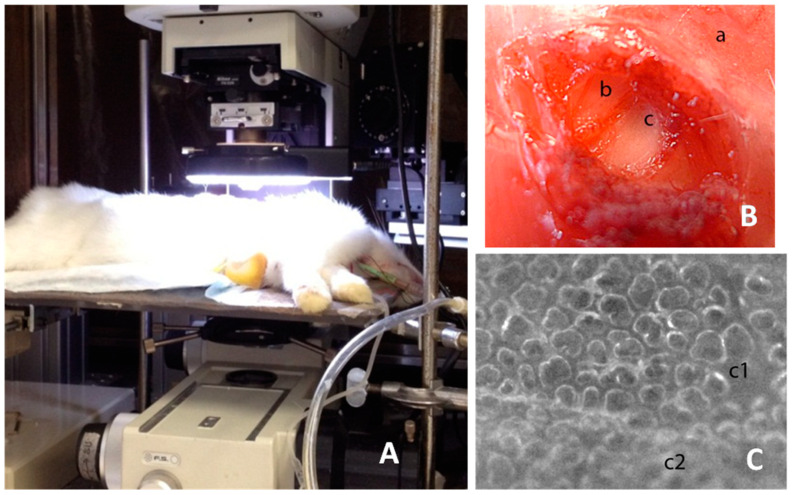
The image shows the experimental setup and the phases to create the pleural window. (**A**) the placement of the animal under the microscope; (**B**) shows the external intercostal layer (a), the internal intercostal layer (b), and the denuded parietal pleura (c); (**C**) shows the portion of the stripped parietal pleura (c1) so as to create the “pleural window”, allowing a neat view of the underlying lung surface (note the difference with the unstripped portion c2). The figure is free of ethical problems and the experimental setup shown is unprotected for publication.

**Figure 2 jimaging-05-00022-f002:**
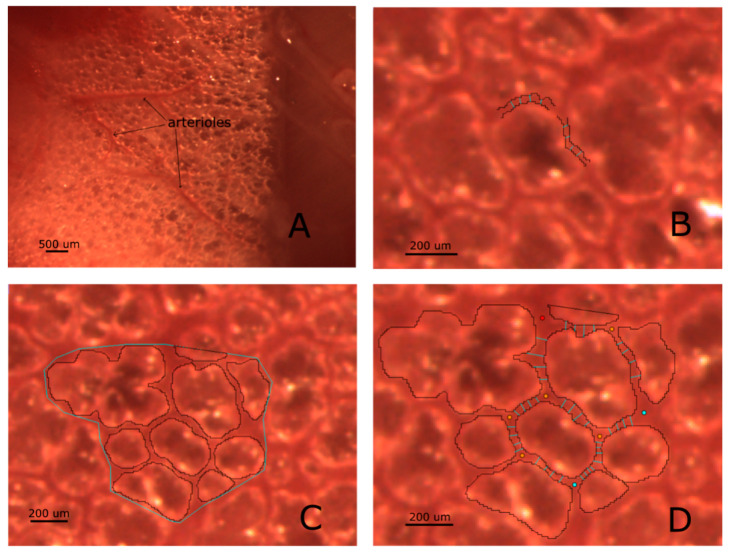
Panel (**A**) is representative of a subpleural region imaged through the intact pleura: alveoli as well as surface microvessels are clearly visible. Panel (**B**) illustrates the result of the method used to derive the diameter of a vessel (we name this technique “segmentation”), based on the definition of microvessels borders described in Methods; vessel diameters are estimated by averaging four adjacent diameters. The same approach is used for both distribution and corner vessels. In Panel (**C**), an estimate of interstitial space is given by the difference between the area encompassing the Region of Interest (ROI) and the overall sum of alveolar surfaces. Panel (**D**) clarifies how a 2D morphological model can be obtained from subpleural images: arteriolar inputs and venular outputs access points are defined (red and cyan points, respectively); corner vessel diameter is estimated by averaging four adjacent diameters.

**Figure 3 jimaging-05-00022-f003:**
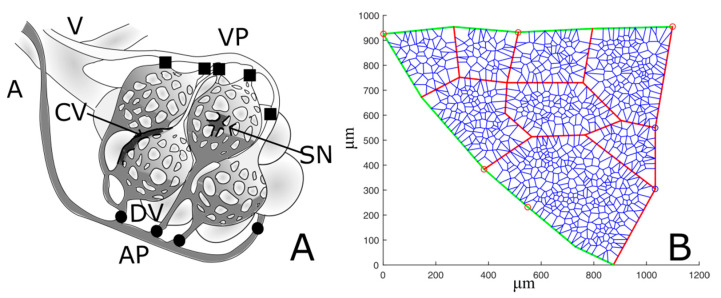
(**A**) simplified schema of lung alveolar capillary units. The arteriole (A) feeds the distribution vessels (DVs) and the arteriolar access points (APs) to the alveolar unit; blood flows into the two districts placed in parallel, the septal network (SN) and the corners vessels (CVs), and finally collects through the venular exit points (VPs) into the venule (V). (**B**) Representative example of an ACU network. Green and red segments represent respectively distribution vessels and corner vessels, while the remaining blue network depicts septal circulation. Red and blue circles identify respectively arteriolar access points and venular exit points.

**Figure 4 jimaging-05-00022-f004:**
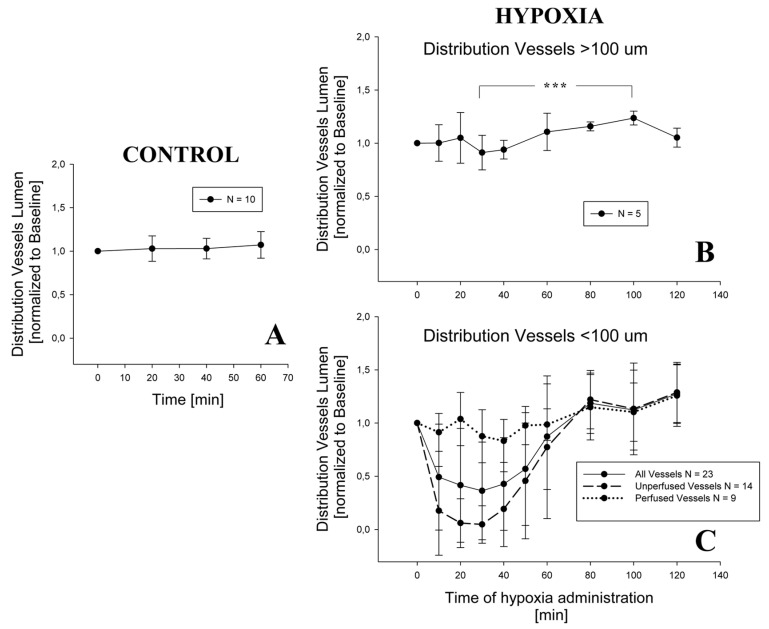
Change in distribution vessels diameter in control condition (Panel **A**) and during hypoxia exposure, both for vessels having a diameter > 100 μm (Panel **B**) and for the smaller ones (Panel **C**). For graphical reasons, the statistical significance is shown only in Panel B (*** = *p* < 0,001), while for Panel C see the text. In Panel C, time patterns are shown for distribution vessels that remain patent during the experiment and for those which close at least at one time point.

**Figure 5 jimaging-05-00022-f005:**
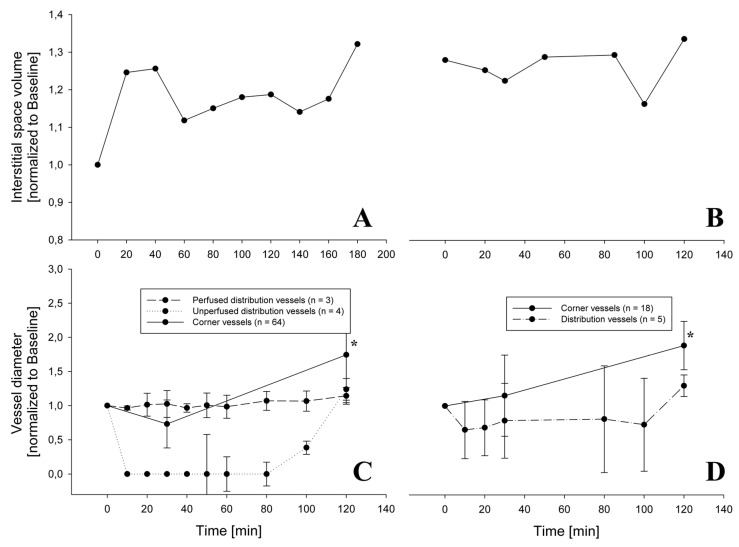
Change of interstitial space volume during hypoxia exposure for 2 different rabbits (Panels **A**,**B**) and the relative change in diameter of distribution and corner vessels (Panels **C**,**D**). In Panel C, distribution vessels have a diameter less than 100 μm and are distinguished among those which were always perfused and those which were closed at some timepoint. In Panel **C**, at all timepoints from 10 up to 100 min, the diameter was significantly decreased relative to baseline for the unperfused distribution vessels (*p*< 0.001). The diameter of corner vessels steadily increased (* = *p* < 0.05).

**Figure 6 jimaging-05-00022-f006:**
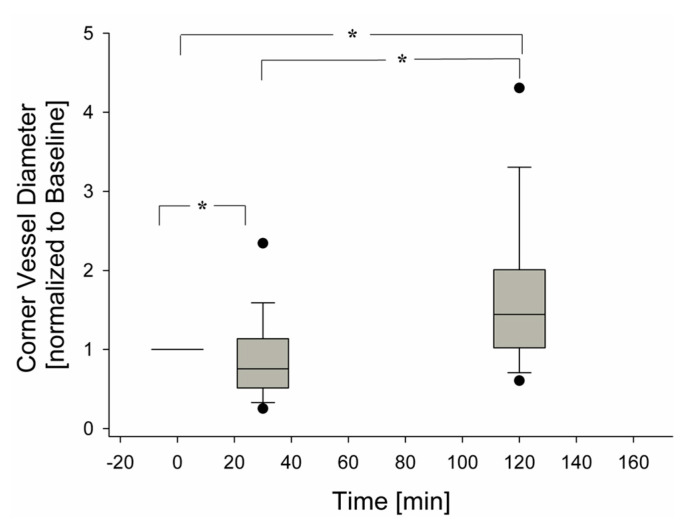
Change of corner vessel diameters over time, during hypoxia exposure (120 observations), (* = *p* < 0.05).

**Figure 7 jimaging-05-00022-f007:**
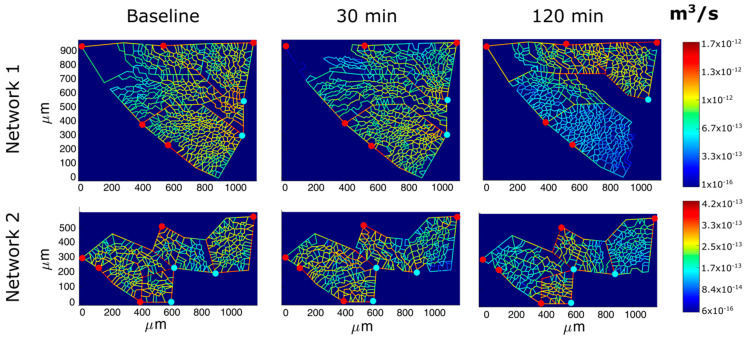
Results of perfusion analysis for two different ACU regions. Red and light blue dots identify respectively arteriolar accesses and venular exits. The color panels show ACU capillary blood flow at different time points (baseline, 30, and 120 min of hypoxia exposure) with color-coded log-scale intensity.

**Table 1 jimaging-05-00022-t001:** This table illustrates the input pressure parameters (cmH_2_O) for the modelled ACU networks. Note the increase in peri-microvascular interstitial pressure (*P_i_*) on developing interstitial edema [3].

Parameter	Baseline	30 min	120 min
*P_art_*	16	16	16
*P_ven_*	6	8	8
*P_i_*	−10	−6	3

**Table 2 jimaging-05-00022-t002:** This table illustrates the main topological and fluid-dynamic parameters for the modeled ACU networks.

Parameter	ACU Network	Baseline	30 min	120 min
Mean Corner Length (um)	1	273 ± 76		
	2	195 ± 58		
ACU Flow (10^−12^ m^3^/s)	1	1.92	1.26 (−34%)	1.78 (−7%)
	2	1.77	1.30 (−27%)	1.46 (−18%)
Septal-to-Corner Flow Ratio	1	3.86	3.56 (−8%)	0.95 (−75%)
	2	0.98	1.84 (+88%)	0.86 (−12%)
Filtration surface (10^−7^ m^2^)	1	7.41	7.21 (−3%)	6.08 (−18%)
	2	3.66	3.88 (+6%)	3.29 (−10%)
Filtration flow/Filtration Surface (10^−10^ m/s)	1	3.54	3.21 (−9%)	5.11 (+44%)
	2	2.50	2.730 (+9%)	3.26 (+30%)

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
