# Peer review of "Understanding Vasomotion of Lung Microcirculation by In Vivo Imaging"

_2313-433X, 2019, doi:10.3390/jimaging5020022_

Reviewer 1 Report

Mazzucca et all. Provide a remarkable technical and scientific advance in imaging the in vivo dynamics of the lung vasculature in physiologic as well as in hypoxic condition.

Specifically, the proposed methodology allows to morphologically discriminate different blood vessel in the context of the lung parenchyma and measure significant changes of the vessel diameters in the context of tissue hypoxia. The observation of a different behaviour between corner and septal vessels confirm both specificity and accuracy of the of the reported observations. Altogether the method and relative results represent a robust scientific and technical platform for an accurate and extensive investigations on the vascular dynamics in lungs.

Minor points

The data described in paragraph 3.1 concerning the Wet-to-dry ratio and arterial PO2 are not reported along the text. The data should be included in dedicated graph or indicated as “data not show”.

The terms “hypoxia administration” should be replaced with “hypoxia exposures”.

Were the differences is statistically significant (pValue<0.05) it need to be included in the relative graph by using the conventional annotation (pValue<0.05 *; <0.005 **; <0.0005***).< span="">

Author Response

We thank the reviewer for the observations and modified the manuscript accordingly. Please find in the following the answers to the specific points addressed by the reviewer.

We noticed that Figure 5 (ex. Fig 4) presented an incorrect graph in Panel A which showed a decrease of the interstitial volume, while in the text we indicated that interstitial volume increased, associated with edema progression. We greatly apologize for the error and we present now the right graph in Panel A.

Answers to Reviewer 

1. We provide both Wet-To-Dry ratio in control conditions and after 120 minutes of hypoxia exposure and the PaO2 values determined in control conditions and after 30 min of hypoxia exposure.

2.Hypoxia administration is replaced with hypoxia exposure.

3.Figures have been modified accordingly, when possible, by inserting the standard legend for statistical significance.

Reviewer 2 Report

This paper studies lung microcirculation in distant vessels under a transient edemagenic condition (hypoxia) in anesthetized rabbits. Microscopic images of distribution vessels and corner vessels were obtained through a pleural window. Manual segmentation of the images provided measures of vessel diameters as well as interstitial space volume. Then, a theoretical model of fluid dynamics was used to estimate microvascular filtration flow in the septal capillary network, which could not be imaged directly. The results show that hypoxia caused initial vasoconstriction followed by reperfusion with regional differences, possibly indicating redistribution of flow, in particular showing that large alveolar units seem less efficient to counteract edema formation.

The manuscript is well structured and well written. It remains concise in part due to references to the authors’ previous publications using similar methods. I think the study is well conducted and interesting.

 I only have minor comments regarding the manuscript:

- In general, some of the vocabulary used was a little weird to me (‘caliper’, ‘distant districts’, ‘general consensus’ obtained from Ethical Committee…), but I do not know if this is common vocabulary in the field that I am unfamiliar with myself, or if this is the result of language barrier. In any case, I would recommend having a native English speaker proofread the manuscript to provide minor improvements in the text fluidity, though it already reads nicely.

- Appendix line 301, space missing between ‘flow’ and ‘across’

- Figure 1 legend is not detailed enough to me. It is not entirely clear what is shown. I think figures shoud be understandable (up to a certain level of detail, of course) on their own.

- Perhaps my most important comment, regarding the manual segmentation of alveolar units. Have the authors estimated the precision or error of their manual segmentation? I am not an expert at lung imaging, but looking at Fig. 1C, the borders are not all that clear to me. It makes me wonder if the determination of borders could be subject to a certain % of uncertainty and how that would affect the results and conclusions of the paper.

- A significant proportion of the references cited are previous publications of the manuscript’s authors. Perhaps a wide literature review including results from different groups would provide a more comprehensive view of the current state of knowledge and how this work complements, refutes or confirms it.

Author Response

We thank the reviewer for the appropriate observations and questions that focus on the most critical aspects of the work. Please find in the following the answers to the specific points addressed by the reviewer.

We noticed that Figure 5 (ex. Fig 4) presented an incorrect graph in Panel A which showed a decrease of the interstitial volume, while in the text we indicated that interstitial volume increased, associated with edema progression. We greatly apologize for the error and we present now the right graph in Panel A.

Answers to the reviewer

- The paper was slightly reviewed to improve text fluidity.

-  Fig.2 (ex Fig.1) now shows a neater image, thus providing a better picture of how alveolar borders are defined. Also the legend of Fig.2 (ex Fig.1) is extended and specific reference is made to the model developed to estimate borders, both for vascular segmentation and for alveolar segmentation. The same point is now developed more extensively in the "Methods" paragraph. We specify that borders are defined using grey levels, thus we use black and white images and not color images, which are anyway shown for greater clarity.

We now provide, as requested, a comprehensive view of the general problem under study by adding references both from our group as well as from others to focus on the state of knowledge and interpretation of results.

Reviewer 3 Report

Review of J. Imaging Manuscript 400396
Title: Understanding vasomotion of lung microcirculation by in vivo imaging

Authors: E. Mazzuca et al.

This in vivo study has reported vascular response in pulmonary microcirculation under edemagenic condition. The vessel networks of alveoli of the anesthetized small animals were longitudinally captured through pleural window with a stereo microscope during 120min-long-hypoxia, and then changes in vessel lumen diameter and blood flow rate, and interstitial volume in the alveoli were measured directly or using image-modelling strategy.

This manuscript can be accepted provided that authors’ answers regarding following comments and questions are satisfactory.

1. In abstract, it is not clear about what is decreased in the sentence: “On the average, hypoxia caused a significant decrease in …”

2. In abstract, there seems to be the missing of suffix for some of phrases: lung extravascular water, in-vivo imaging the micro-vascular morphology.

3. In 2.1 Experimental Setup, please add an another figure showing the imaging set-up involving rabbits under the experiment.

4. Please shortly describe how to make this window on the chest of rabbit in this manuscript.

5. Please explain how the rabbits were exposed to a hypoxic mixture. And please mention the evidence for relevance of hypoxia to pulmonary edema with references.

6. In Fig. 1, shape of contours around alveoli in (c) are different from those in (d) for the same location. Did you make sketch these boundaries manually?

7. In Fig. 1, how could authors’ estimate the caliper of microvessels such as corner vessels from the microscope image? It looks very difficult to see the vessels with naked eyes. Please indicate the distribution vessels and corner vessels in images in Fig. 1.

8. In subsection of Estimate of interstitial space volume, what does the ‘grey level’ of this sentence “…grey level between air and tissue phase (Fig. 1C)” mean for? I think the microscope image captured is not gray-scaled but color CMOS image. And please mark the boundary between air and tissue phase for readers not used to the figure.

9. In Fig. 2(b), to my knowledge, partitioning of planes in Voronoi diagram is based on set of points designated. I am wondering how authors could display the septal circulation (blue lines) in the Voronoi image. Did you maybe manually set many of points in random in ROI?

10. In the caption of Fig. 6, “red segments represent respectively distribution vessels and corner vessels…” The distribution and corner vessels are all red-colored? And indication of red and blue circles is not recognizable in this figure because of blue background.

11. In Fig. 6, the blood flow rates of networks at 120 min appear more decreased compared to 30 min but authors claimed the progressive increase of ACU flow from table 2 (e.g., 1.26 at 30min to 1.78 at 120min). Please explain this.

Author Response

We thank the reviewer for the appropriate observations and questions that focus on the most critical aspects of the work and we modified the manuscript accordingly. Please find in the following the answers to the specific points addressed by the reviewer.

We noticed that Figure 5 (ex. Fig 4) presented an incorrect graph in Panel A which showed a decrease of the interstitial volume, while in the text we indicated that interstitial volume increased, associated with edema progression. We greatly apologize for the error and we present now the right graph in Panel A.

Answers to the reviewer

Points 1, 2) Abstract is rewritten according to the provided indications.

Point 3,4) The experimental set-up was shown in a new figure (Fig. 1) which shows the animal preparation and the imaging set-up; also details of surgical preparation are now provided.

Point 5) Details of hypoxia exposure are given, including reason for the edemagenic role of hypoxia and references            

Points 6, 7, 8) Fig.2 (ex Fig.1) now shows a neater image, thus providing a better picture of how alveolar borders are defined. Also the legend of Fig.2 (ex Fig.1) is extended and specific reference is made to the model developed to estimate borders, both for vascular segmentation and for alveolar segmentation. The same point is now developed more extensively in the methods paragraph. We specify that borders are defined using grey levels, thus we use black and white images and not color images, which are anyway shown for greater clarity.

Point 9) Relative to the Voronoi diagram, a better explanation is given: "to build the septal capillary network, we chose a random distribution of N Voronoi points within the ACU. The value of N was chosen so as to match the morphological constraint of the ratio between capillary and interstitial alveolar volume equal to 1.54, a value provided for rabbits ([10]) corresponding to the maximum extension of a pulmonary capillary network". 

Point 10) Fig. 7 (ex Fig. 6) was improved in readability by increasing dot size. The phrase "red segments represent respectively distribution vessels and corner vessels" was a typo and has been removed; we apologize for it and thank the reviewer for having noticed it.

Point 11) Relative to the question of why blood flow rates of networks at 120 min appear more decreased compared to 30 min but a progressive increase of ACU flow is shown in table 2, the answer is: ACU flow in network 1 is compromised in a large part of the network, but the lower resistance in the upper right part largely contributes to the higher flow. This point has been also added to the text.

Round  2

Reviewer 3 Report

Authors have made much efforts to refine and polish many of contents in the original manuscript according to the comments of reviewer. Therefore, the revised manuscript became convincing and sound in reporting their work. Hence, I would like to recommend acceptance in present form.

Author Response

We thank the reviewer for appreciating the effort for our revised manuscript.